# PSTO: Learning Energy-Efficient Locomotion for Quadruped Robots

**Wangshu Zhu** 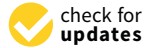 **and Andre Rosendo ***

Living Machines Laboratory, School of Information Science and Technology, ShanghaiTech University, Shanghai 201210, China; zhuwsh@shanghaitech.edu.cn
* Correspondence: arosendo@shanghaitech.edu.cn

**Abstract:** Energy efficiency is critical for the locomotion of quadruped robots. However, energy efficiency values found in simulations do not transfer adequately to the real world. To address this issue, we present a novel method, named Policy Search Transfer Optimization (PSTO), which combines deep reinforcement learning and optimization to create energy-efficient locomotion for quadruped robots in the real world. The deep reinforcement learning and policy search process are performed by the TD3 algorithm and the policy is transferred to the open-loop control trajectory further optimized by numerical methods, and conducted on the robot in the real world. In order to ensure the high uniformity of the simulation results and the behavior of the hardware platform, we introduce and validate the accurate model in simulation including consistent size and fine-tuning parameters. We then validate those results with real-world experiments on the quadruped robot Ant by executing dynamic walking gaits with different leg lengths and numbers of amplifications. We analyze the results and show that our methods can outperform the control method provided by the state-of-the-art policy search algorithm TD3 and sinusoid function on both energy efficiency and speed.

**Keywords:** machine learning; robot locomotion; energy efficiency; deep reinforcement learning

## 1. Introduction

Legged locomotion [1] is essential for robots to traverse difficult environments with agility and grace. However, the energy efficiency of mobile robots still have room for improvement when performing a dynamic locomotion. Classical approaches often require extensive experience of the structure and massive manual tuning of parameteric choices [2,3].

Recently, learning-based approaches, especially deep reinforcement learning methods, have achieved tremendous progress in controlling robots [4–7]. Policy search [8], as a subfield of deep reinforcement learning, is widely studied in recent years. Numbers of policy search algorithms have appeared to improve the performance, sample efficiency while reducing the entropy in the learning process e.g., DDPG [4], TRPO [5], PPO [9], SAC [10], and TD3 [11]. These algorithms automate the training process and produce feasible locomotion for robots without much human interference.

While these methods have demonstrated promising results in simulation, the transfer of those to the real world often performs poorly, including substandard energy efficiency and low speed, which is mainly caused by the reality gap [12]. Model discrepancies between the simulated and the real physical system, unmodeled dynamics, wrong simulation parameters, and numerical errors contribute to this gap. In three-dimensional locomotion, this gap will be even amplified because the subtle difference of the contact situations between the simulation and the real world could be magnified and forked to unexpected consequences. With this gap, robots could conduct poor performance, increase energy consumption, and even damage themselves. Works on narrowing the reality gap are essential for machine learning on robots.

On the other hand, studies on learning directly in the real world have been conducted recently. In robot grasping and robot locomotion, these studies have achieved gratifying results [13,14]. Compared with grasping, learning in robot locomotion in the real world is more challenging due to the difficulties of automatically resetting the environments and continuously collecting data [15]. Additionally, a robot falling can potentially damage joints, legs, or torso, which will affect the learning result. Therefore, learning in simulation is more reasonable and efficient when the robot is expensive and fragile. After that, a well-designed policy could be transferred to the real robot and the robot can perform locomotion learned from simulation, e.g., [16–18].

Power is a very limited resource in robots, especially for untethered walking robots. Researchers have made great efforts on improving the energy efficiency of walking robots. Adaptive gait pattern control was introduced to establish a stable locomotion for quadrupedal robots while suppressing the energy consumption [19]. A gradient descent based optimal control approach was utilized to optimize the energy cost of a biologically inspired three-joint robot [20]. The influence of parameters that defined the foot trajectories of legged robots was investigated to minimize energy consumption [21]. The pulsed control signal outperforms the sinusoidal counterpart with higher energy efficiency on a hopping robot [22]. Some other papers studied the hardware parts e.g., structure of the robots and backdrivable actuator designs [23,24] to improve energy efficiency because energy is critical for the long-term behavior of robots. The quadrupedal robot that can morphologically adapt to different environmental conditions has also been carried out to achieve higher energy efficiency for different terrains [25].

To conduct massive parameter tuning, population optimization include black-box [26] and Bayesian optimization [27] is an appealing approach. Bayesian optimization is often data efficient, so it is practical to apply it on real robots directly [28,29]. However, these methods are challenging to scale up to high-dimensional control space especially in the real world.

Inspired by the insights above, in this paper, we present a complete learning and optimizing approach for quadruped robot locomotion, named Policy Search Transfer Optimization (PSTO), which unites the strengths of both deep reinforcement learning and optimization. In our approach, the energy-efficient control policy is learned in simulation and transferred to the real robot after optimization.

There are two main challenges in our approach: learning robust control policy and transferring the policy to the real physical system. To learn a robust control policy, we adopt TD3 as the training method in simulation, which is outstanding in preventing overestimation and can obtain a state-of-the-art control policy within limited training time. To better transfer the policy to the real world, we first unify the robot size in simulation and in the real world. Additionally, we improve the fidelity of the physics simulator by adding an actuator model with appropriate gear parameters. In addition, we further optimize the policy from the simulation by numerical methods before conducting it on the real robot. We evaluate our approach on a quadruped robot Ant with two different groups of leg length. We show our method can generate an energy-efficient locomotion method. We can successfully transfer controlling policies to the real robot with an accurate physics simulator and optimization. Furthermore, we investigate the power consumption related to speed on the Ant robot, which can provide a useful guideline for readers to construct their robots with energy-efficient locomotion.

## 2. Materials and Methods

### 2.1. Physics Simulation and Robot Platform

The training environment is a Pybullet [30] based simulation platform, Tianshou [31], in which we build a physics simulation of the Ant robot (Figure 1 Left). Pybullet solves the equations of motion for rigid bodies in the system while satisfying physical constraints involving joint limits, earth contact, and actuator models. Tianshou is a reinforcement

learning platform, which provides a fast and parallelizable framework with Python to build a deep reinforcement learning agent.

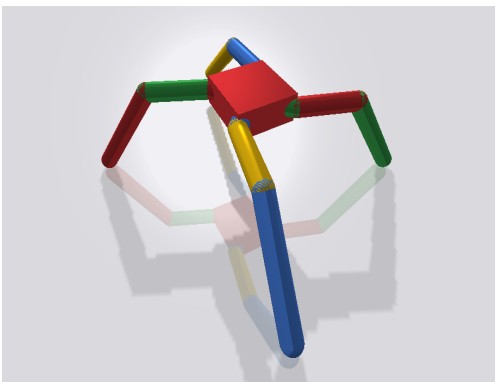 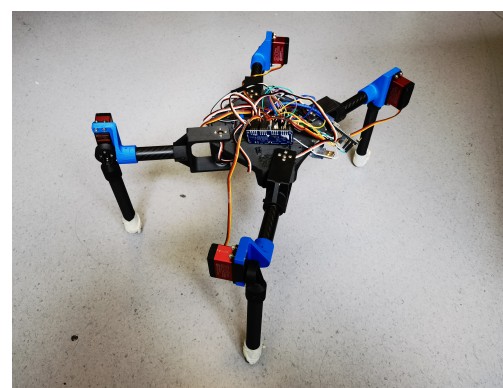

**Figure 1.** The Ant robot in Pybullet simulation (**Left**) and in the real world (**Right**) are created at a consistent size to guarantee the performance in the real world.

The robot platform we used is the Ant [29], a custom-built quadruped robot with eight servomotors (Figure 1 Right). Each leg of the Ant consists of two carbon fiber tubes and two servo motors controlled by an Arduino nano with an Adafruit 16-Channel PWM/Servo Driver. Servomotors on the hips are responsible for the movement of the legs in the horizontal plane, while servomotors on the knees are responsible for the movement of the legs in the vertical plane. Four hip joints move in the same horizontal plane to avoid oscillations in the region below the body. The range of each servomotor is limited to $[-45°, 45°]$ and the control frequency is 60 Hz. The snapshots of the Ant robot walking on the marked arena with PSTO and leg length of 1 L (23 cm) are illustrated in Figure 2. We unify the robot size in simulation and in the real world to guarantee performance in the real world. In addition, we improve the fidelity of the physics simulator by meticulously tuning the parameters of the robot in the simulation, including gear and motor control range.

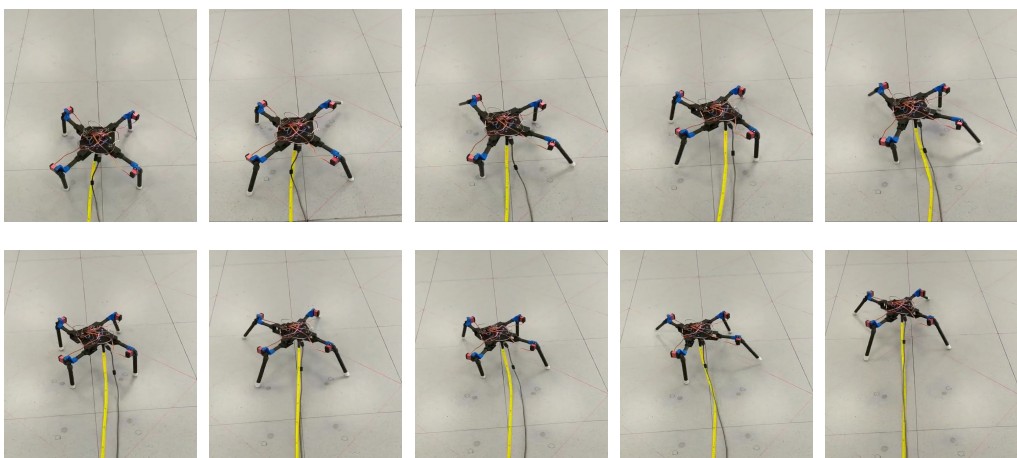

**Figure 2.** Snapshots of the Ant robot walking on the marked arena within 12 s with PSTO and leg length of 1 L (23 cm). The side length of each black square on the ground is 1 m.

We measure the energy efficiency of the robot system calculating its cost of transport (CoT) [32], with

$$CoT = \frac{E}{Wd} \tag{1}$$

where $E$ is the energy consumption of the robot system during the whole locomotion, $W$ is the weight of the robot, and $d$ is the horizontal translating distance. In this paper, we use

INA219 DC current sensors to measure the voltage and current of the system. The energy consumption of each time step is integral to the final energy consumption $E$. Usually, we consider that the system with lower CoT is more energy-efficient.

## 2.2. Deep Reinforcement Learning

We consider reinforcement learning (RL) as a process running by a Markov decision process (MDP). An MDP is defined by the tuple $(\mathcal{S}, \mathcal{A}, p, \rho_0, r, \gamma)$, where $\mathcal{S} \subseteq \mathbb{R}^{d_s}$ is a finite set of states of the agent and $\mathcal{A} \subseteq \mathbb{R}^{d_A}$ is a finite set of actions which can be conducted by the agent. $p(s_{t+1} \mid s_t, a_t)$ denotes the probability of transfer from state $s_t$ to $s_{t+1}$ given the action conducted by the agent at time step $t$. $p(s_{t+1} \mid s_t, a_t)$ is usually unknown in high-dimensional control tasks, but we can sample from $p(s_{t+1} \mid s_t, a_t)$ in simulation or from a physical system in the real world. $\rho_0$ denotes the initial state distribution of the agent, and $r(s_t, a_t)$ denotes the valued reward in state $s_t$ when the agent executes action $a_t$. The last $\gamma \in (0, 1)$ is the discount factor to indicate that recent rewards count more. A policy in RL is defined as the map (or function) from the state space to the action space. In policy search, the goal is to find a optimal policy $\pi$ that maximizes the long-term expected reward (accumulated reward) [5],

$$\eta(\pi) = \mathbb{E}_{\rho_\pi(s)\pi(a|s)}[Q^\pi(s, a)] \tag{2}$$

where

$$Q^\pi(s_t, a_t) = \mathbb{E}_{s_{t+1}, a_{t+1}, \dots}\left[\sum_{t=0}^{T} \gamma^t r(s_t, a_t)\right] \tag{3}$$

in which $\rho_\pi(s)$ is the state distribution under $\pi$, i.e., the probability of visiting state $s$ under $\pi$. $Q^\pi(s_t, a_t)$ is called state–action value function which evaluate the state–action pair. $s_{t+1} \sim p(s_{t+1} \mid s_t, a_t)$, $a_t \sim \pi(a_t \mid s_t)$ and T is the terminal time step. In recent works, the policy usually adopts the deep neural network structure (e.g., [4,5,9]), which simulates the network of neurons of human beings to some extent and can conduct backpropagation to update the network.

The original ant model in Pybullet has 111 dimensions of observations, including positions of the torso in Cartesian coordinates, orientation, joint angles, 3D velocities, angular velocities, joint velocities, and external forces. However, we exclude the $x, y$ positions in Cartesian coordinates and external forces to reduce the dimension of observation to 27 for faster training. We do not exclude the $z$-position because this value is critical for the Ant to know whether it will fall and how to behave in high and low positions. We find that our observation space is sufficient to learn the walking policy for the Ant. For the action space, we select the position control mode of the actuators for better transferring to Arduino-controlled servomotors. The reward function includes alive reward, moving progress, electricity cost, joints at limit cost, and feet collision cost, which encourages the robot to move faster to the target position while avoiding high energy consumption and collisions as shown in:

$$r(s_t, a_t) = r_{alive} + v_{fwd} - c_e - c_{limit} - c_{collision} \tag{4}$$

where $r_{alive}$ is the alive reward which is 1 when the distance between the body of the robot and ground is larger than 0.05 m, and otherwise it is $-1$ at each time step. $v_{fwd}$ is the forward velocity of the robot. $c_e$ is the electricity cost of the robot. $c_{limit}$ is the joints at limit cost which discourages stuck joints. $c_{collision}$ is the feet collision cost that prevents the robot from smashing its feet into itself.

We adopt TD3 [11] as our training method, which applies clipped double Q-learning, delayed update of target and policy networks, and target policy smoothing on DDPG [4] to prevent the overestimation of the value function. The learning curves are shown in Figure 3 and we will further discuss the training detail in Section 2.4.

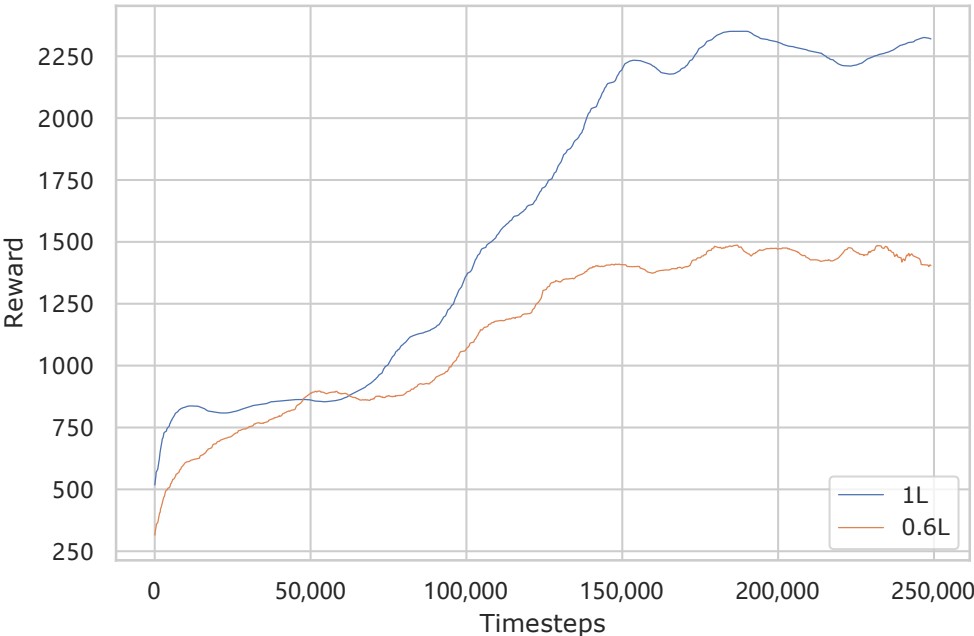

**Figure 3.** Reward on each time step on a 1 L (23 cm) and 0.6 L (13.8 cm) leg length simulation. The training process shows that longer legs reach higher rewards in the later learning stages.

### 2.3. Trajectory Optimization

The direct transfer from the simulation to the real world is running the policy learned from the simulation on the real robot [28]. However, in this control scheme, the sensor provides the observations to the policy and the policy gives back the action (control command) to the actuators, which could cause large latency between the computed action and the current state. In addition, the robot needs to have a system to perform the neural network for each time step, which could waste some computing sources. An alternative to conducting the transfer is generating a feasible trajectory by the simulation for the real robots. Trajectory control is widely utilized in industrial robots, which guarantee the safety and stability of robots.

With the robust policy in the simulation, we can produce a simulation trajectory for each joint of the Ant by averaging several trials as shown in Figure 4. In this paper, we average 20 trials to provide the original trajectory. However, even considering the power penalty in the simulation, there is the reality gap between the simulation and the real world, which is mainly caused by the model discrepancies between the simulated and the real physical system. In addition, the highly dynamic motion also increases the uncertainty of the robot during locomotion. The robot can only move sub-optimally along with the original trajectory.

One insight on the original trajectory is that it has a large bias. The possible reason for that is that each isolated angle value does not consider the nearby values in the recent time steps. Besides the average method, we adopt polynomial regression to further address this problem. Polynomial regression is a widely used regression method with one independent variable which can be expressed as:

$$q_i = \beta_0 + \beta_0 x_i + \beta_2 x_i^2 + ... + \beta_k x_i^k + e_i \tag{5}$$

where $q_i$ is the output independent variable, which is the optimized trajectory in this paper, and $x_i$ is the input variable, which is the original trajectory. Each $\beta_k$ is the slope of the regression surface with respect to variable $x_i$, the random error of the $i$th case is $e_i$, and $k$ is the degree (order) of the polynomial. In this paper, we choose $k = 10$ as the regression degree to provide smooth trajectories while avoiding overfitting, as shown in Figure 4. In

addition, we amplify the trajectories to investigate the energy efficiency of the robot, which will be further discussed in Section 2.4.

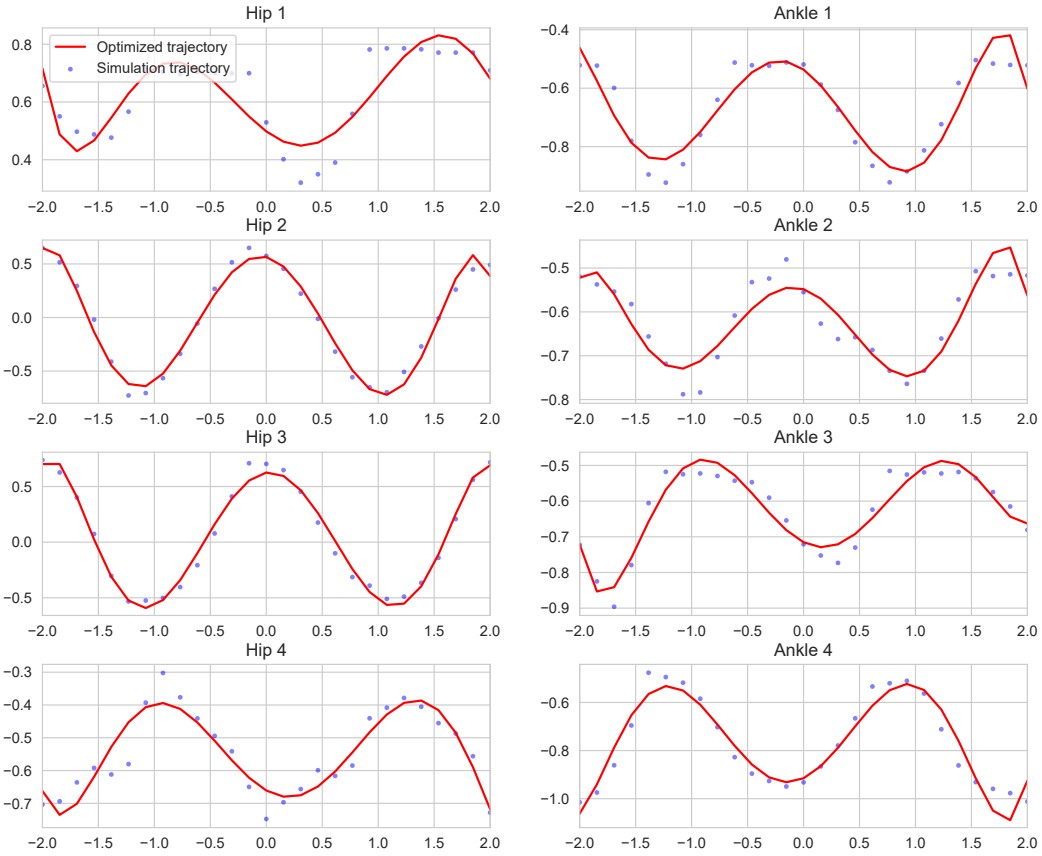

**Figure 4.** The blue dots are the angle trajectories of eight joints over time (s) on a simulation. The red lines are the trajectories acquired from the optimization, and each loop takes 4 s. Simulated trajectories have a larger variance, while the optimized ones are smoother, which leads to a higher energy efficiency on the real robot. We later compare this smoothness with sinusoidal curves.

With the aforementioned insights in mind, we propose a novel approach named Policy Search Transfer Optimization (PSTO), as shown in Algorithm 1, which integrates deep reinforcement learning and optimization to provide a energy-efficient control method for real robots.

---

**Algorithm 1** PSTO with TD3.

---

**Input:** The robot model in simulation
1: Initialize critic networks $Q_{\theta_1}$, $Q_{\theta_2}$, and actor network $\pi_\phi$ with random parameters $\theta_1$, $\theta_2$, $\phi$
2: Initialize target networks $\theta_1' \leftarrow \theta_1$, $\theta_2' \leftarrow \theta_2$, $\phi' \leftarrow \phi$ and replay buffer $\mathcal{B}$
3: **for** $t = 0 \rightarrow T$ **do**
4:     Select action with exploration noise $a \sim \pi(s) + \epsilon$
5:     Observe reward $r$ and new state $s$
6:     Store transition tuple $(s, a, r, s')$ in $\mathcal{B}$
7:     Update critics parameters $\theta_1$, $\theta_2$ by TD3
8:     Delayed update policy networks $\phi$ and target networks $\theta_1'$, $\theta_2'$, $\phi'$ when $t$ mod $d == 0$
9: **end for**
10: Record the trajectory $q_i$ for each joint from the simulation
11: Optimize each trajectory $q_i$ by polynomial regression and amplification
**Output:** Optimized trajectory for the real robot

---

*2.4. Experiment Setup*

We present experiments to assess the performance of our proposed algorithm on the Ant robot with two groups of leg length and three methods. We compare our method with control methods obtained from the aforementioned state-of-the-art TD3 algorithm and the well-known sinusoid control function to evaluate the energy efficiency of different methods. In addition, for each trajectory from the method, we conduct eight different amplitudes ranging from $1.0 \times$ to $2.4 \times$ to further investigate the energy efficiency of different gaits.

We first conduct the training process in the Pybullet-based training platform Tianshou on an Ant model with 1 L (23 cm) leg length and 0.6 L (13.8 cm) leg length. We create a new Ant model with consistent leg length, gear, and motor control range with the real robot. Our training networks include two actor networks and two critic networks as recommended in TD3. The actor and critic networks have three fully connected hidden layers and 128 neurons per layer, and the adopted activation function is ReLU function [33]. The algorithm runs with ten random seeds, 100 epochs, and 2500 steps per epoch using ADAM optimizer [34]. The other training hyperparameters are shown in Table 1 for more details, and the learning curves are shown in Figure 3. The 1 L leg length will achieve a higher reward than 0.6 L leg length after about 100,000 time steps though 0.6 L leg length has a slightly higher reward around 50,000 time steps. The training time is about 4.8 h on the DELL OptiPlex 7060 series desktop with i7-8700 processor, 12 processor threads, and 32 GB internal memory.

**Table 1.** Hyperparameter of the training environment.

| Hyperparameter | Value |
|---|---|
| buffer size | 20,000 |
| actor learning rate | 0.0003 |
| critic learning rate | 0.001 |
| $\gamma$ | 0.99 |
| tau | 0.005 |
| update actor frequency | 2 |
| policy noise | 0.2 |
| exploration noise | 0.1 |
| epoch | 2500 |
| step per epoch | 100 |
| collect per step | 10 |
| batch size | 128 |
| hidden layer number | 3 |
| hidden neurons per layer | 128 |
| training number | 8 |
| test number | 50 |
| joint gear | 45 |
| joint control range | $-45 \sim 45°$ |

Among the ten random seeds, we choose the policy with the best reward to provide trajectories in simulation. The policy will perform a walking gait, and we average 20 trials to generate eight original trajectories as the blue dots in Figure 4 for four hip actuators and four ankle actuators. After that, we apply a 10-degree polynomial regression method from sklearn library on the trajectories to produce smoother trajectories for the real robot. The midpoints of the trajectory of hip 2 and hip 3 are near 0 while other joints are not, which indicates that we can obtain asymmetric trajectories to operate the robot and most of the trajectories have subtle phase differences from the training process. Our optimization method can also change the flat part of the trajectory to some curves that can provide more energy-efficient locomotion for the robot.

The sinusoid function is widely used in open-loop control, offering advantages in simplicity and robustness. We also choose the sinusoid control function obtained from Bayesian optimization [29] as one of the comparisons of our method. In addition, we

increase the amplitude of the trajectories from three methods, ranging from 1.0× to 2.4×, to evaluate the energy efficiency of the robot.

## 3. Results

In this section, we demonstrate the comparison of original, sinusoid, and PSTO functions on energy efficiency and speed in real-world experiments. Figure 5 presents a comparison between three methods on our 1 L leg length robot moving at a 0.12 m/s speed. The average power consumption per step of our proposed PSTO method is 0.776 Watts, while the original and sinusoidal trajectories consumed 0.837 and 0.802 Watts, respectively. In this initial real-world experiment, our proposed method could save 7% energy at the 0.12 m/s speed compared to the original and sinusoidal trajectories.

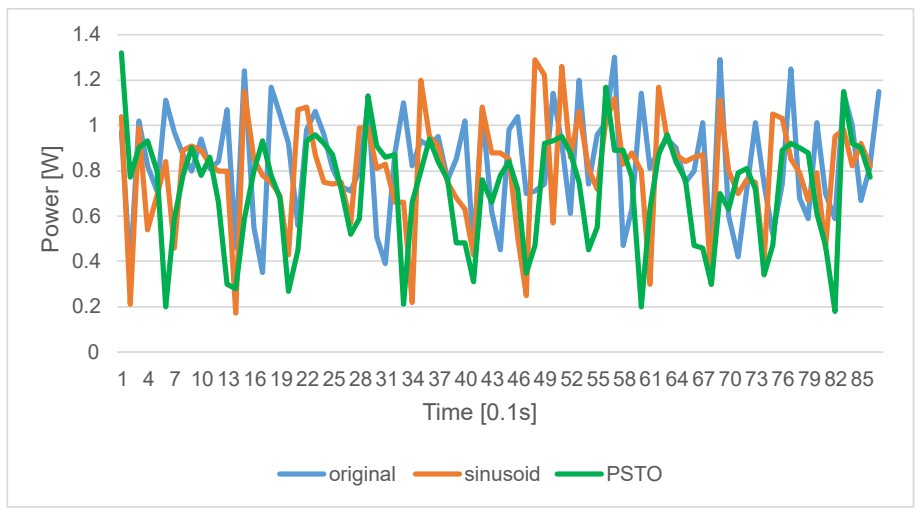

**Figure 5.** The comparison of power consumption per time step of three methods in the real world. PSTO has a lower average power consumption than both original and sinusoid functions when the speed is 0.12 m/s and leg length is 1 L.

For each trial, the running time is 12 s, which is reasonable for the robot to perform several gait cycles. Figure 6 shows the CoT related to amplification ranging from 1.0× to 2.4× with the interval of 0.2, with both 1 L and 0.6 L leg lengths. The best CoT from PSTO reaches 0.595 when the amplification is 2.0, which is 14% lower than the other two methods with the same amplification and the leg length of 1 L in consideration.

While the sinusoid method shows a high efficiency at low amplifications when short legs are used, PSTO outperforms all methods with mid to high amplification. PSTO can save up to 38% energy when compared to the original method and 26% if compared to the sinusoid method with a motion range amplification of 2.2×.

Figure 7 shows speed changes in relation to the adopted amplification. The highest speed reached by PSTO is 0.139 m/s for both 1 L leg length and 0.6 L leg length, which is 39% higher than the other two methods for 1 L leg length and 18% higher for 0.6 L leg length. The highest speed appears with an amplification of 2.0× for 1 L leg length while the amplification is 2.4× for 0.6 L. A possible reason for the similarity in maximum speed for 1 L and 0.6 L is that the larger amplification leads to an unstable locomotion with longer legs due to the higher center of mass. This instability causes excessive slipping, which eventually decreases the speed of the robot. Since the original trajectory has a larger variance, which would exceed the control range of the actuator with a large amplification, the robot can not conduct the experiment with high amplification following the original trajectory. Independent of amplification, the best CoT and speed reached by those three methods can be seen in details in Table 2. PSTO can achieve a 15% lower CoT and 10% higher speed than the original and sinusoid methods for 1 L leg length, and it can reach a 26% lower CoT and 30% higher speed than the other two methods with 0.6 L leg length.

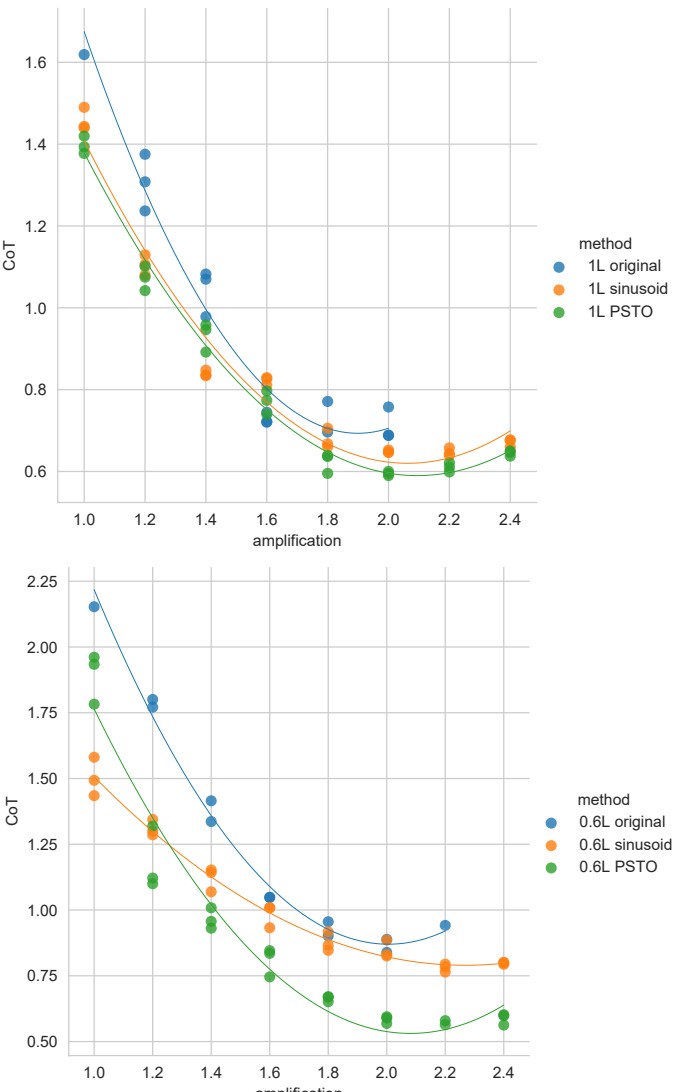

**Figure 6.** CoT related to amplification for original trajectory, sinusoid trajectory, and PSTO trajectory for 1 L leg length (**top**) and 0.6 L leg length (**bottom**) in the real-world experiments. The amplification range is from 1.0 to 2.4. The curves are the 2nd order polynomial regression for the tendency of three methods. PSTO can achieve lower CoT cost than the other two methods for different amplifications when the leg length is 1 L. For 0.6 L leg length, PSTO has competitive energy efficiency among the three methods when the amplification is low. With the increasing amplification, PSTO can achieve outstanding energy efficiency. Notice that the original trajectory has a larger variance which will exceed the control range of the actuator with a large amplification, so the robot can not conduct the full experiment following the original trajectory.

**Table 2.** Comparisons of original method, sinusoid method and PSTO with 1 L leg length and 0.6 L leg length in the real-world experiments.

| Method | CoT | Speed [m/s] |
|:---:|:---:|:---:|
| 1 L original | 0.688 | 0.127 |
| 1 L sinusoid | 0.639 | 0.127 |
| 1 L PSTO | 0.595 | 0.139 |
| 0.6 L original | 0.839 | 0.096 |
| 0.6 L sinusoid | 0.764 | 0.104 |
| 0.6 L PSTO | 0.563 | 0.139 |

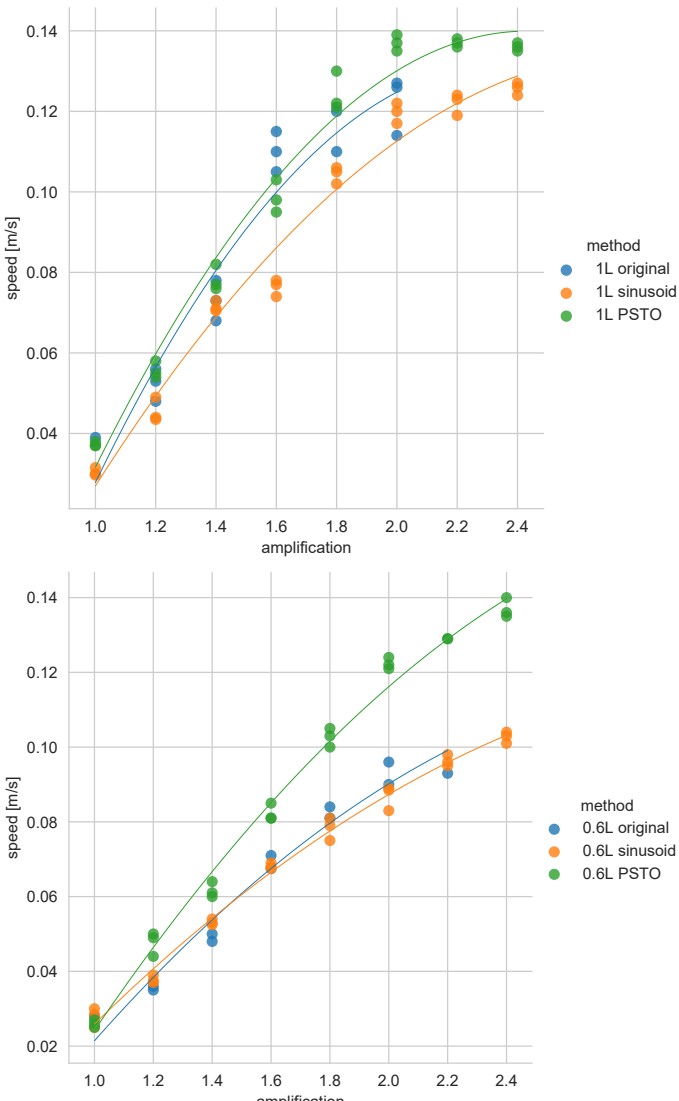

**Figure 7.** Speed [m/s] related to amplification for the original, sinusoid, and PSTO trajectories for 1 L leg length (**top**) and 0.6 L leg length (**bottom**) in the real-world experiments. The amplification range is from 1.0 to 2.4. The curves are the 2nd order polynomial regression for the tendency of three methods. PSTO can achieve a higher speed than the other two methods for different amplifications for both 1 L and 0.6 L leg length. Notice that the original trajectory has a larger variance which will exceed the control range of the actuator with a large amplification, so the robot can not conduct the full experiment following the original trajectory.

In addition, we analyze the CoT in relation to speed, as shown in Figure 8, which could be a reasonable measurement of energy efficiency. During low-speed locomotion, the sinusoid method has better performance than the other two methods as it shows a lower CoT with the same speed for both leg lengths. However, PSTO starts to outperform the sinusoid method when the robot reaches higher speeds. PSTO can achieve a 20% higher energy efficiency than the sinusoid method and a 30% higher efficiency when compared to the original method with speeds above 0.12 m/s.

The crossing point of PSTO over sinusoid appeared earlier when the leg length is 0.6 L. The possible reason for that is that PSTO works better when the system needs a more meticulous control method due to the optimization process. We also observe that most of the tendency curves appear to converge with the increasing speed, which means that the speed limitation of the system could be about 0.139 m/s.

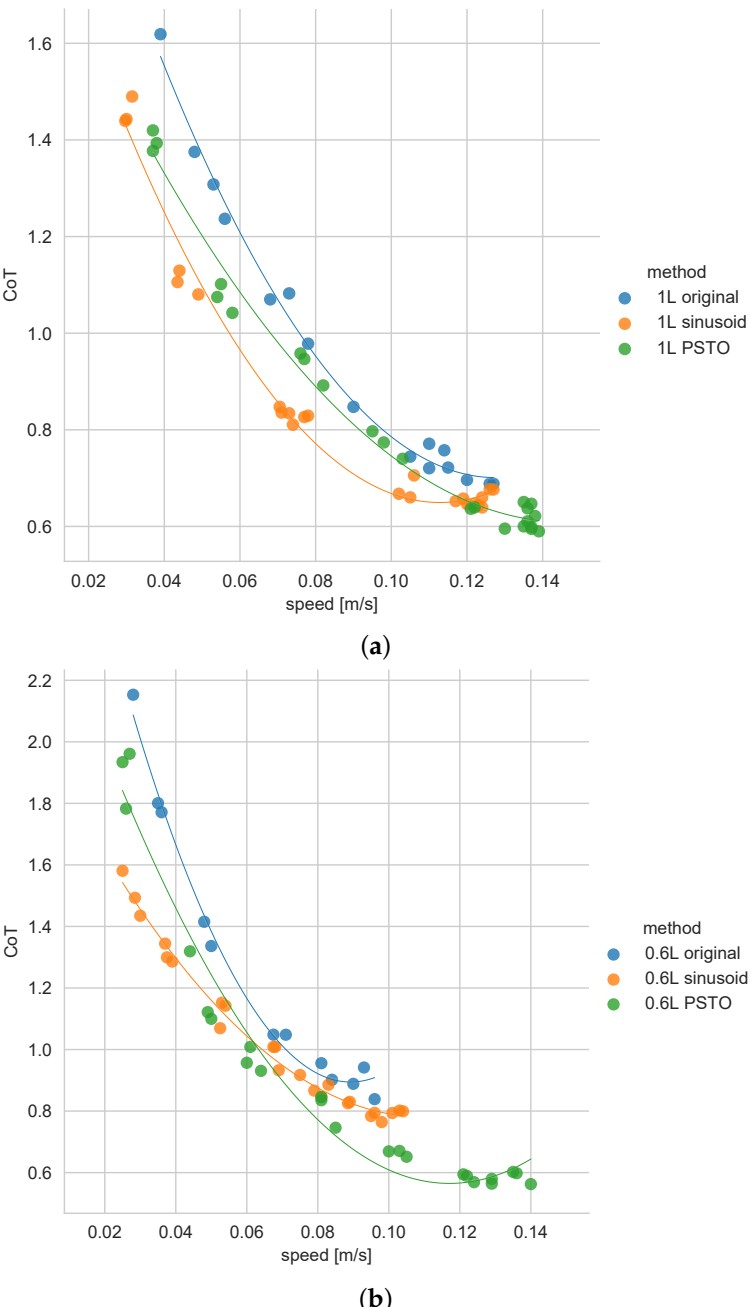

**Figure 8.** CoT related to speed [m/s] for original trajectory, sinusoid trajectory, and PSTO with trajectory amplification from 1.0 to 2.4 for 1 L leg length (**a**) and 0.6 L leg length (**b**) in the real-world experiments. The curves are the 2nd order polynomial regression for the tendency of three methods. At low speed, the sinusoid method is more competitive with low CoT than other methods while PSTO can achieve higher speed and lower CoT with the increasing amplification for both 1 L and 0.6 L leg length.

## 4. Discussion

Compared to a previously published hopping robot [22], our Ant robot has a higher CoT with the same speed, which could be caused by the non-backdrivable servomotors, while the hopping robot is equipped with parallel elastic actuators. It is also very likely that the elastic energy stored at the springs helps their system move efficiently, but the detrimental effects of that previously proposed actuation can be the lack of control, sudden speed changes, and even directional changes. Backdrivable servomotors could further

improve the energy efficiency of leg robots while the control method should be more subtle and robust.

Other than training directly in the real world [29], our method can avoid the potential damage to the robots during the training process which is critical for valuable robots. Additionally, parallel learning with multiple agents and environments can greatly accelerate the training in the simulation.

However, the legged robots are not as energy-efficient as the wheeled robots in general [32], due to the lower friction of the wheels. On the other hand, the morphology of robots can contribute to the energy efficiency to a great extent [19,25], so we will apply our methods on robots with different morphology to further improve the energy efficiency of robots.

## 5. Conclusions

In this paper, we propose a novel method to learn and transfer an energy-efficient control method for a quadruped robot in the real world by deep reinforcement learning and optimization. We use the TD3 algorithm to train the policy for safety and learning efficiency and optimize the control trajectories for energy-efficient locomotion by polynomial regression. We compare our method with the sinusoid method and to trajectories obtained directly from simulation on both energy consumption and speed. We find that our method can outperform the other two methods on the real robot for different groups of leg length. Such findings will contribute to the study on the algorithm design of machine learning, structure design, including length of legs and control range of joints, and locomotion of robots.

While in this paper we focused on the robot Ant, in the future, we will consider other types of robots, which will further expand the applicability of this method. We will also study the relationship between the terrain and structure of robots on energy efficiency and speed because the ability for robots to traverse difficult environments with energy-efficiency and agility is crucial in the real world. Although we study the relationship between the amplitude of the trajectory and the performance of the robot with different leg lengths, there are still many parameters remaining to be explored in the framework.

**Author Contributions:** Conceptualization, W.Z. and A.R.; Data Curation, W.Z.; Funding Acquisition, A.R.; Investigation, W.Z.; Methodology, W.Z.; Software, W.Z.; Supervision, A.R.; Writing—Original Draft, W.Z.; Writing—Review and Editing, A.R. All authors have read and agreed to the published version of the manuscript.

**Funding:** This work was funded by the National Natural Science Foundation of China Grant No. 61850410527 and the Shanghai Young Oriental Scholars Grant No. 0830000081.

**Institutional Review Board Statement:** Not applicable.

**Informed Consent Statement:** Not applicable.

**Data Availability Statement:** The repository with the code materials and the data required to reproduce the results are available at https://github.com/zhuwangshu/PSTO (accessed on 11 December 2021).

**Conflicts of Interest:** The authors declare no conflict of interest.

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
