# Peer review of "PSTO: Learning Energy-Efficient Locomotion for Quadruped Robots"

_machines, doi:10.3390/machines10030185_

Round 1

Reviewer 1 Report

The main objective of this work is to improve the energy efficiency for the locomotion of quadruped robots. The author used a deep reinforcement learning approach (TD3 algorithm) to learn a control policy, and considered the reality gap between the simulation and the real world, so a numerical optimization approach (polynomial regression) was used to generate a smoother open-loop control trajectory for an ant robot in the real world. And the author designed a comparison experiment with sinusoid control function method to demonstrate the superiority of the proposed method in terms of energy efficiency on the real robot.

However, I believe that the innovation of the method in this paper is seriously lacking, and the proposed scheme for integrating reinforcement learning and trajectory optimization is very simple. This is because that the author only use very well-established reinforcement learning algorithms TD3 and common trajectory smoothing methods, polynomial regression.

In addition, the authors proposed to use the optimization method to directly generate an open-loop locomotion trajectory in order to transfer the control strategy trained in the simulation to the real robot. The motion control strategy, although effective on the real robot, completely discards the ability to walk adaptively in the environment, because the robot is unable to make changes in its actions based on the feedback state. In other words, the robot can only walk on a simple terrain that has been designed. Therefore, I believe that the proposed method does not have any practical significance.

In summary, the authors should consider designing more innovative approaches, and provide more insights in reinforcement learning training or simulation-to-reality transfer. Furthermore, it would be necessary and valuable for the robot to achieve adaptive locomotion in varying environments.

Reviewer 2 Report

The main contribution of this paper is the PSTO algorithm which combines deep reinforcement learning and optimization. The robot is first trained in simulation and

then transferred to the real world. The optimization is performed by numerical methods and executed on the robot in the real world. Results are validated in simulation and the robot in the real world. 

To improve this paper the authors should include a discussion of other energy efficient gaits in addition to the sinusoidal.

The authors should compare the energy consumption with other models, such as.

Bing, Z., Lemke, C., Cheng, L., Huang, K., & Knoll, A. (2020). Energy-efficient and damage-recovery slithering gait design for a snake-like robot based on reinforcement learning and inverse reinforcement learning. Neural Networks, 129, 323-333. Nygaard, T. F., Martin, C. P., Torresen, J., Glette, K., & Howard, D. (2021). Real-world embodied AI through a morphologically adaptive quadruped robot. Nature Machine Intelligence, 3(5), 410-419. Tsujita, K., Tsuchiya, K., & Onat, A. (2001, November). Adaptive gait pattern control of a quadruped locomotion robot. In Proceedings 2001 IEEE/RSJ International Conference on Intelligent Robots and Systems. Expanding the Societal Role of Robotics in the the Next Millennium (Cat. No. 01CH37180) (Vol. 4, pp. 2318-2325). IEEE.   Also the authors should explain the results more clearly. Please add the gait data. Please thoroughly check the use of references (sometimes space is added before them, and sometimes, not. Always add space between text and reference).  

Reviewer 3 Report

The article presents the method  energy-efficient control method for a quadruped robot. The topic of the article is actual for the low energy efficiency of walking robots. 
To verify the method, it is necessary to improve the methodology of the experiment because they are not clear to me. 
It is necessary to describe the kinematic model of a walking robot.
It is necessary to provide basic data in order to verify the experiment. Authors don't properly show the data.

Round 2

Reviewer 3 Report

The authors of the article incorporated comments and described the methodology. They provided basic data on the measured values for verification. After the inspection, I did not find any discrepancies. The authors also described the kinematics and types of actuators used to build the robot. I recommend publishing the article in this form.